# ACTIVE PROBABILISTIC CLUSTERING

## ABSTRACT

Active Constrained Clustering (ACC) is a widely used semi-supervised clustering framework to improve clustering quality through progressive annotation of informative pairwise constraints. However, the application of existing ACC methods to large datasets with numerous classes incurs high computational or query expenses. In this paper, we analyze the inefficiency of pair-based and sample-based ACC and the rationale behind cluster-based ACC. Moreover, we provide the theoretical guarantee for cluster fusion under a certain purity constraint and a clustering quality constraint with respect to normalized mutual information (NMI). Drawing on these theoretical insights, we introduce a novel Active Probabilistic Clustering (APC) framework designed to scale effectively with large datasets. Compared to previous methods, APC demonstrates superior performance across eight datasets of varying sizes (ranging from 350 to 100,000 samples) in terms of clustering quality, query cost, and computational expense. Specifically, APC accomplishes satisfactory clustering outcomes (e.g., NMI > 0.95) using 3,920 queries on a dataset with 100,000 samples, while baseline methods yield inferior clustering results (e.g., NMI ≤ 0.85) with 10,000 queries. Concurrently, APC operates at a speed 100x faster than baseline methods.

## 1 INTRODUCTION

Active Constrained Clustering has been extensively investigated in various semi-supervised clustering scenarios, such as person re-identification (Zhao et al., 2013) and plant species identification (Kumar et al., 2012; Xiong et al., 2016). It is particularly useful for handling datasets with a vast and unknown label space that would require expert knowledge for ground truth class identification (Basu et al., 2004; Mai et al., 2013; Xiong et al., 2016). This is because ACC simplifies the labeling task by only requiring the oracles to judge whether two samples belong to the same class or not (must-link or cannot-link), which is feasible even for untrained non-expert humans. The query results are utilized as the constraints for the downstream constrained clustering part. ACC improves the performance of semi-supervised clustering by incorporating these pairwise constraints in a carefully selected manner to guide the constrained clustering process. Therefore, selecting informative sample pairs to query is critical for the success of ACC.

Existing ACC methods are in three major categories with pair-based, sample-based, and cluster-based query strategies, respectively (Xiong et al., 2016). **(1) Pair-based methods** (Xu et al., 2005; Wauthier et al., 2012; Abin, 2017) select informative sample pairs from a total of $\binom{N}{2}$ pairs, where $N$ represent the dataset size. These methods typically suffer from high computation costs due to the iterative measurement of $O(N^2)$ pairs. **(2) Sample-based approaches** (Basu et al., 2004; Xiong et al., 2016) reduce the cost by building neighborhoods with the selected samples, where different neighborhoods consist of samples queried to belong to different classes (Xiong et al., 2013). Nevertheless, these methods encounter challenges when applied to large datasets with numerous classes, as they require at least $\binom{K-1}{2}$ queries to build neighborhoods covering $K$ underlying classes. **(3) Cluster-based methods** (Van Craenendonck et al., 2017; 2018; Shi et al., 2020) reduce the number of queries by querying pairs of cluster centroids obtained from initial clustering results using classical methods such as K-means (Ahmed et al., 2020). However, centroid-based cluster merging may degrade the clustering quality.

This work aims to pursue a cluster-based ACC algorithm with more accurate cluster fusion. Our contributions are summarized below.

**Our Contributions.** We first *provide a theoretical result* to guide the cluster fusion. In specific, we demonstrate that clustering quality with be enhanced if the cluster pair satisfies a 'purity constraint' and the overall clustering quality is sufficient with respect to a 'NMI value constraint'. Intuitively, the purity constraint ensures that the majority of samples within the cluster belong to the same class. The NMI value constraint guarantees a high level of coherence between the current clustering and the real class distribution. Based on this theoretical result, we *propose a new cluster-based ACC framework* named active probabilistic clustering. In particular, APC leverages the fast probabilistic clustering (FPC) algorithm (Liu et al., 2022) for obtaining the initial clustering results. FPC is chosen due to its robustness to noise samples and unknown cluster numbers, as well as its ability to provide pairwise posterior probabilities between sample pairs, facilitating the selection of query samples. Our strategy encompasses the following three ingredients:

- APC selects informative cluster pairs that maximize the NMI gain, where we derive two novel estimators (Section 3.1) of NMI gain associated with candidate queries based on the pairwise posterior probability from FPC.

- APC then examines selected clusters through a 'Human Test' mechanism (Section 3.2), which first examines the purity of these clusters according to our theoretical result (Theorem 1). If the purity constraint is satisfied, it further calls the human query to examine whether the dominant classes of the cluster pair are identical.

- APC then decides whether to merge, split, or not operate the queried cluster pairs according to the examination results. In terms of merging or splitting operations, we propose a fast relabeling strategy to adjust the cluster identity of the samples in the cluster pairs (Section 3.3).

We conduct extensive experiments on eight image datasets ranging from 1,000 to 100,000 in scale. In all cases, APC significantly improves clustering quality with at most a few thousand annotations. In contrast, baselines require thousands of annotations to achieve limited improvements even on datasets in the 1,000-scale range, which demonstrates the superiority of APC on larger datasets.

## 2 CLUSTER-BASED ACTIVE CONSTRAINED CLUSTERING

**Definition 2.1** (ACC). *We denote the true classes of $N$ samples $X = \{x_1, \cdots, x_N\}$ by $Y = \{y_1, \cdots, y_N\}$, where $y_i \in \{1, \cdots, K\}$ and $K$ is the number of classes. ACC aims to improve the clustering performance generated by a clustering algorithm with the supervisory information acquired by human queries. It requires humans to judge if $y_i = y_j$ for some carefully selected sample pairs $\{(x_i, x_j)\}$ and then perform constrained clustering with the constraints, i.e., the query results. ACC differs from conventional semi-supervised clustering methods in that it iteratively selects sample pairs based on some strategy instead of randomly selecting them.*

Effective ACC frameworks should possess three key properties: (1) Monotonicity: users should be able to terminate the process at any point and receive an improved clustering result; (2) Speed: the selection of sample pairs should be swift, minimizing wait times between queries; (3) Efficiency: the framework should require as few queries as possible to achieve a substantial improvement in clustering accuracy(Van Craenendonck et al., 2018). Most pair-based ACC methods fail to meet these requirements, and their high computational complexity poses challenges for scalability. Sample-based ACC methods are more computationally efficient but struggle to fulfill the Efficiency requirement. They confront an Accumulation Plane dilemma when faced with a large number of classes. In this scenario, the clustering performance stagnates, and substantial improvement is not observed until a significant number of queries are executed (see Figure 2).

To pursue effective ACC, recent work has begun exploring the cluster-based ACC framework. This approach replaces the complex constrained clustering part of ACC with some relabeling strategies(Shi et al., 2020) which decides how clusters should be merged or split. However, existing methods lack a definitive criterion to guide the merging process. They require the oracle to query the central sample of two clusters and merge them abruptly if the central samples belong to the same class. This strategy carries risk concerning the Monotonicity property, especially when the purity of some clusters in the clustering is low, which makes the central sample not representative.

In light of this, we introduce a pivotal theorem that offers clear guidance for merging actions. This is achieved through an evaluation of the normalized mutual information (Vinh et al., 2009), which is a measure of how much information two clusters share.

**Definition 2.2** (NMI). *Given $N$ samples and their two clustering $\Omega = \{w_1, \cdots, w_k\}$ and $\Omega' = \{w'_1, \cdots, w'_k\}$, the NMI $n$ between $\Omega$ and $\Omega'$ is defined as*

$$n = \frac{2\mathbb{I}(\Omega; \Omega')}{\mathbb{H}(\Omega) + \mathbb{H}(\Omega')} = \frac{2\mathbb{I}(\zeta; \zeta')}{\mathbb{H}(\zeta) + \mathbb{H}(\zeta')},$$

*where $\zeta = (|w_1|/N, \cdots, |w_k|/N)$ and $\zeta' = (|w'_1|/N, \cdots, |w'_{k'}|/N)$ are two distributions induced by $\Omega$ and $\Omega'$, respectively. Here, $\mathbb{I}(\zeta; \zeta') = \sum_{x \in \zeta, y \in \zeta'} \mathbb{P}(x, y) \log \frac{\mathbb{P}(x,y)}{\mathbb{P}(x)\mathbb{P}(y)}$ denotes mutual information, and $\mathbb{H}(\zeta) = -\sum_{x \in \zeta} \mathbb{P}(x) \log(\mathbb{P}(x)$ denotes entropy.*

**Theorem 1** (Guarantee for Cluster Fusion). *Assume that the clustering of $N$ samples is $\Omega = \{w_1, \cdots, w_k\}$, the ground truth clustering is $C = \{c_1, \cdots, c_K\}$, and the NMI value of $\Omega$ with respect to $C$ is $n_1$. We define the dominant class of a cluster $w$ as $\arg\max_j |w \cap c_j|$, and the purity of $w$ as $\max_j \frac{|w \cap c_j|}{|w|}$. For any two clusters in $\Omega$, say $w_1$ and $w_2$, suppose they have a common dominant class $c_1$ with purity of $t_1, t_2 \in [0.7, 1]$, respectively. After merging $w_1$ and $w_2$ into a new cluster $w_{1,2}$ and obtaining a new clustering $\Omega^\star = \{w_{1,2}, w_3, \cdots, w_k\}$ with NMI value $n_2$, if $n_1 \geq 2 \cdot (1.0586 - \min\{t_1, t_2\})$, then we have $n_2 \geq n_1$.*

Theorem 1 offers guidance for the merge operation in cluster-based ACC, and the detailed proof is in Appendix A.1. Specifically, merging two classes can achieve provable benefits when their purity is at least $0.7$ and previous NMI exceeds the threshold $2 \cdot (1.0586 - \min\{t_1, t_2\})$. Notably, these two constraints are generally satisfied when the clustering is based on features extracted with deep neural networks (e.g., Liu et al., 2023). Therefore, we propose that a more effective cluster-based ACC framework should aim to achieve maximum improvement in NMI with a minimal number of queries. To fulfill this objective, we introduce Active Probabilistic Clustering (APC).

## 3 ACTIVE PROBABILISTIC CLUSTERING

### 3.1 QUERY STRATEGY

In this subsection, we introduce our query strategy, which includes the formulation of a metric for assessing the impact of a human query—specifically, merging clusters when their dominant classes are identical. Inspired by Theorem 1, we propose to estimate the expected improvement of NMI for this query operation:

$$\mathbb{E}[\Delta\text{NMI} \mid w_i, w_j] = \mathbb{P}(w_i = w_j \mid w_i, w_j) \cdot (n_2 - n_1), \tag{1}$$

where $n_1$ and $n_2$ are the original NMI and updated NMI after merging $w_i$ and $w_j$, respectively. For the conditional probability expressions within, the condition $w_i$ implies that the samples within this cluster share a common class, and the same interpretation applies to the condition $w_j$. $w_i = w_j$ signifies the possible observed event that the classes of the two clusters are identical. In what follows, we present how to approximate $\mathbb{P}(w_i = w_j \mid w_i, w_j)$ and $(n_2 - n_1)$.

Liu et al. (2022) establishes a probabilistic clustering framework that characterizes the clustering state of samples using fundamental variables $e_{st}$, where $e_{st} = 1/0$ signifies whether $y_s$ equals $y_t$ or not. The framework assumes independence among different $e_{st}$, an assumption we uphold throughout this paper. As $(w_i = w_j | w_i, w_j)$ equals $(\forall s \in w_i, \forall t \in w_j, e_{st} = 1 \mid \forall (s, t) \in w_i \text{ or } w_j, e_{st} = 1)$, we can estimate the conditional merging probability $\mathbb{P}(w_i = w_j \mid w_i, w_j)$ with $\mathbb{P}(e_{st} = 1)$. The procedure of estimating pairwise probability matrix $P_{N \times N} = \{P(e_{st} = 1)\}_{s,t \in [1, \cdots, N]}$ is given in Appendix B.1. Subsequently, leveraging the independence assumption and the conditional probability formula, we can express the merge probability as follows:

$$\mathbb{P}(w_i = w_j \mid w_i, w_j) = \frac{\prod_{s \in w_i, t \in w_j} \mathbb{P}(e_{st} = 1)}{\prod_{s \in w_i, t \in w_j} \mathbb{P}(e_{st} = 1) + \prod_{s \in w_i, t \in w_j} \mathbb{P}(e_{st} = 0)}. \tag{2}$$

The detailed derivation of Eq. (2) is in Appendix A.2. However, Eq. (2) is relatively sensitive to the estimation error of $\mathbb{P}(e_{st} = 1)$ when recalling cluster pairs, as it focuses on all pairwise relationships.

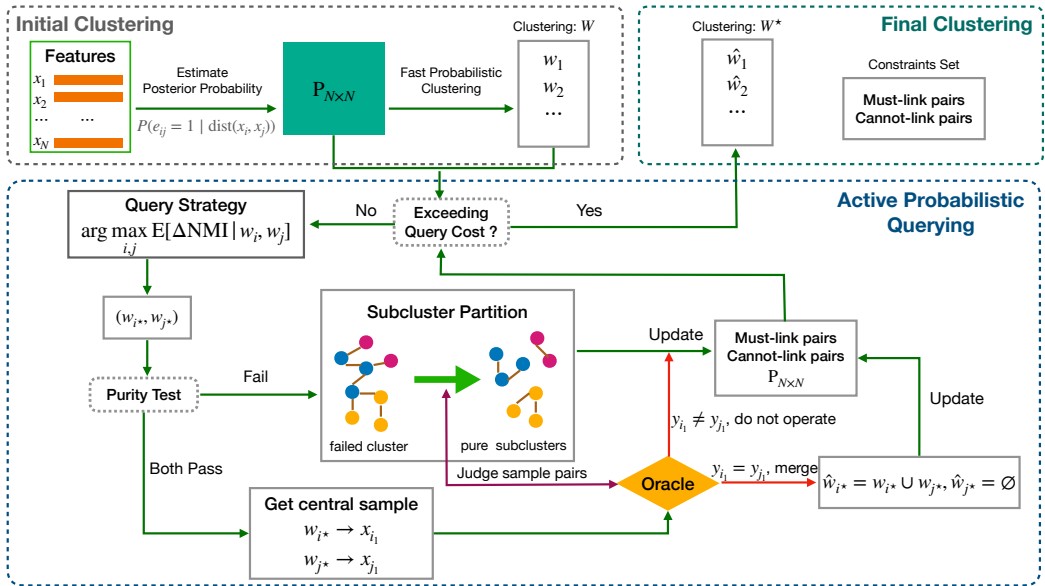

Figure 1: The workflow of Active Probabilistic Clustering (APC) methods comprises three phases. In the **Initial Clustering Phase**, we employ the Fast Probabilistic Clustering algorithm to obtain the input for APC, which includes the posterior probability matrix $\mathrm{P}_{N \times N}$ and clustering result $W$. In the **Active Probabilistic Querying Phase**, APC iteratively selects a cluster pair using a query strategy. Subsequently, it conducts a Purity Test on both clusters. If both clusters pass the test, APC prompts the oracle to query their central samples to determine whether these clusters should be merged. If at least one cluster fails the Purity Test, APC initiates a Subcluster Partition process, wherein oracles assess sample pairs in the failed cluster, leading to its division into several pure subclusters. APC then updates the constraints set and the $\mathrm{P}_{N \times N}$. This cycle continues until a predefined query cost is reached. In the **Final Clustering Phase**, APC outputs the new clustering result $W_{\star}$.

Specifically, the pairwise probability between inlier and outlier samples from $w_i$ and $w_j$ will degrade the recall probability $\mathbb{P}(w_i = w_j | w_i, w_j)$ (i.e., making the result biased towards 0). To deal with this issue, (reciprocal) k-nearest neighbors are widely used in clustering (Liu et al., 2023). Inspired by this, we refine the merge probability as follows:

$$\mathbb{P}_{\times}(w_i = w_j \mid w_i, w_j) = \frac{\prod_{s \in w_i, t \in \mathrm{knn}_{w_j}(s)} \mathbb{P}(e_{st} = 1)}{\prod_{s \in w_i, t \in \mathrm{knn}_{w_j}(s)} \mathbb{P}(e_{st} = 1) + \prod_{s \in w_i, t \in \mathrm{knn}_{w_j}(s)} \mathbb{P}(e_{st} = 0)}, \quad (3)$$

where $\mathrm{knn}_{w_j}(s)$ denotes the $k$-nearest neighbors of $s$ in $w_j$. Additionally, as an alternative to $\mathbb{P}_{\times}$, we introduce the following $\mathbb{P}_{+}$, which is more computationally efficient and robust to noises:

$$\mathbb{P}_{+}(w_i = w_j \mid w_i, w_j) = \frac{\sum_{s \in w_i, t \in \mathrm{knn}_{w_j}(s)} \mathbb{P}(e_{st} = 1)}{\sum_{s \in w_i, t \in \mathrm{knn}_{w_j}(s)} \mathbb{P}(e_{st} = 1) + \sum_{s \in w_i, t \in \mathrm{knn}_{w_j}(s)} \mathbb{P}(e_{st} = 0)}. \quad (4)$$

We will use both $\mathbb{P}_{\times}$ and $\mathbb{P}_{+}$ in the experiments where they achieve comparable performance. Also, we compare them to the query strategy in previous cluster-based ACC methods and find that $\mathbb{P}_{+}$ shows a significant advantage in large datasets. See Section 4.3 for more discussions.

Now we consider how to approximate $n_2 - n_1$. Following the notations in Theorem 1, we denote $\Delta h = \mathbb{H}(\Omega) - \mathbb{H}(\Omega^{\star})$ and have the approximation: $n_2 = \frac{2\mathbb{I}(\Omega^{\star};C)}{\mathbb{H}(\Omega^{\star})+\mathbb{H}(C)} \approx \frac{2\mathbb{I}(\Omega;C)}{\mathbb{H}(\Omega)+\mathbb{H}(C)-\Delta h}$, where we use the fact that $\mathbb{I}(\Omega^{\star};C) \approx \mathbb{I}(\Omega, C)$ when the purity of $w_i$ and $w_j$ is sufficiently large ($\geq 0.7$). Moreover, when the sizes of clusters $w_i$ and $w_j$ are significantly smaller than the sample size $N$, the direct calculation gives that $\Delta h \ll \mathbb{H}(\Omega) + \mathbb{H}(C)$. Refer to Appendix A.3 for verification. Hence, we have $n_2 - n_1 = \frac{2\mathbb{I}(\Omega;C)\Delta h}{(\mathbb{H}(\Omega)+\mathbb{H}(C)-\Delta h)(\mathbb{H}(\Omega)+\mathbb{H}(C))} \approx \frac{2\mathbb{I}(\Omega;C)\Delta h}{(\mathbb{H}(\Omega)+\mathbb{H}(C))^2} \propto \Delta h$. Combing this with Eq. (1), Eq. (3), and Eq. (4), we obtain our estimators of $\mathbb{E}[\Delta \mathrm{NMI} \mid w_i, w_j]$:

$$\mathbb{E}[\Delta \mathrm{NMI} \mid w_i, w_j] \propto \mathbb{P}_{\times}(w_i = w_j \mid w_i, w_j)\Delta h \quad \text{or} \quad \mathbb{P}_{+}(w_i = w_j \mid w_i, w_j)\Delta h. \quad (5)$$

To ensure a high success rate in establishing "must-link" connections among recalled cluster pairs, We employ a two-step query strategy: (1) we filter out low-quality cluster pairs by retaining those whose conditional merge probability surpasses a predefined threshold denoted as $\mathbb{P}_l$; (2) we employ Eq. (5) to compute the expected NMI gain for each cluster pair for ranking and selection.

## 3.2 HUMAN TEST

In this subsection, our primary target is to ascertain whether the selected cluster pairs should be merged. To achieve this goal, we introduce an assessment mechanism referred to as the 'Human Test'. This test encompasses two essential steps: (1) conducting a Purity Test on the chosen clusters to evaluate if they satisfy the 'purity constraint'; (2) selecting the representative sample for each cluster and requiring the oracles to judge if they belong to the same class. The details are as follows:

**Purity Test.** Given that the ground truth purity is unattainable without knowledge of their class information, we alternate it by estimating the extent to which samples within a cluster are densely concentrated in the probability space. We quantify the test on a cluster $w$ as follows:

$$\mathcal{PT}(w) = \mathbf{1}\Big(\frac{\sum_{i \in w, j \in w(i)} \mathbb{P}(e_{ij} = 1)}{\sum_{i \in w} |w(i)|} > \tau\Big), \quad w(i) = \{j \mid j \in w, \mathbb{P}(e_{ij} = 1) < \mathbb{P}(e_{ij_{\mathrm{mid}}} = 1)\}, \quad (6)$$

where $j_{\mathrm{mid}}$ refers to the sample in cluster $w$ that holds the intermediate position in terms of distance from sample $i$. and $\tau$ is a predefined default threshold (e.g., 0.7). More implementation details are presented in Appendix B.2.

**Human Query.** If both clusters pass the purity test, we select the central sample from them and require the oracle to judge if the classes of the samples in this pair are identical. In probabilistic clustering, the central sample of a cluster $w$ is defined as $\arg\max_i l(i) = \sum_{j \in w, j \neq i} \log \frac{\mathbb{P}(e_{ij}=1)}{\mathbb{P}(e_{ij}=0)}$.

## 3.3 CLUSTERING UPDATE

In this subsection, we discuss updating the clustering with the queried constraints. We first introduce our relabeling strategy which adjusts the cluster label of the cluster pair, then presents how to update the constraints set and the pairwise probability matrix $\mathrm{P}_{N \times N}$.

**Relabeling Strategy.** For different outcomes in the 'Human Test', we have that (i) In cases where the Purity Test result is 1 for both clusters, if the Human Query result indicates a must-link, we promptly assign the same label to all samples in the cluster pair (merge them into a single cluster); otherwise, we retain the original clustering labels; (2) If the Purity Test result is 0 for one of the clusters, we need to split it into several new subclusters, each with higher purity. Considering the efficiency of sample-based ACC in dealing with small-scale datasets, we directly employ it for this partition task and describe the procedure in Algorithm 2.

**Transitive Inference.** The must-link and cannot-link constraints possess the transitivity property (e.g., $(x_1, x_2), (x_2, x_3)$ are must-linked, then $(x_1, x_3)$ is must-linked). To store the constraints, we define a state matrix as $S = \{s_{ij}\}_{N \times N}, s_{ij} \in \{-1, 0, 1\}$. Here, 1/-1 denotes must-link/cannot-link, and 0 indicates an unqueried state. To avoid unnecessary queries, we need to augment the constraints set each time a new constraint is added. We assert that this expansion is only relevant to the preceding constraints that share a common sample with the new constraint, and propose a Fast Transitive Inference (Algorithm 3) method to update the constraints. The correctness of this assertion and algorithm is proved in Appendix B.3. Meanwhile, for the sample pairs in the constraints set, we update their pairwise probability by $\mathbb{P}(e_{st} = 1) = 1 - \epsilon$ for must-link pairs, and $\mathbb{P}(e_{st} = 1) = \epsilon$ for cannot-link pairs, where $\epsilon = e^{-4}$.

`APC Framework`. Overall, we summarize the workflow of the Active Probabilistic Clustering (APC) algorithm (Algorithm 1) in Figure 1. Specifically, we adopt the Fast Probabilistic Clustering (FPC) algorithm (Liu et al., 2022) to generate our initial clustering result for two considerations: (1) FPC does not require any ground truth information; (2) FPC isolates noise samples into outlier clusters, leading to high purity in the remaining clusters, typically exceeding 0.7 in practice.

## 4 EXPERIMENTS

We organize the experiments as follows: We first introduce our experiment settings in Section 4.1; explore how effective is the proposed APC as compared to baseline ACC methods in Section 4.2; Then we discuss how the proposed query strategy influence the effectiveness of APC, especially in alleviating the category fission problem in Section 4.3; Finally, we investigate whether APC can handle large datasets more effectively than other baseline methods in Section 4.3.

**Algorithm 1:** Active Probabilistic Clustering

**Input:** threshold $\mathbb{P}_l$, $\tau$, query limit $Q_{\max}$, queries
  $q = 0$
Obtaining $\mathrm{P}_{N \times N}$ and initial clustering $\Omega$ with FPC
**for** *epoch in* $1 : T$ **do**
  Get candidate cluster pairs as
  $\mathcal{C} = \{(w_i, w_j) | w_i, w_j \in \Omega, \mathbb{P}_+(w_i = w_j) > \mathbb{P}_l\}$
  Calculate and rank the cluster pairs in $\mathcal{C}$ with
  Eq. (5)
  **while** $\mathcal{C}$ *is not empty and* $q < Q_{\max}$ **do**
    Remove the first pair from $\mathcal{C}$ as $(w_1, w_2)$
    Implement Purity Test on $w_1$ and $w_2$
    **if** *Both clusters pass the test* **then**
      Select central samples $x_1$ and $x_2$ from
        $w_1$ and $w_2$ respectively
      Require oracle to query $(x_1, x_2)$
      Merge $w_1$ and $w_2$ if $(x_1, x_2)$ is
        must-linked
    **else**
      Split $w_1$ or/and $w_2$ with Algorithm 2
    **end if**
    Update constraints set with Algorithm 3
    Update $q$ by adding the newly invested
      number of queries
  **end while**
**end for**

**Algorithm 2:** Subcluster Partition

**Input:** cluster $w$; subclusters $\mathcal{N} = \{\}$
Sort w in descending order with $l(i)$
**for** $i$ *in* $w$ **do**
  Select the cloest sample from each
    subcluster in $\mathcal{N}$, and get $J$
  Query $i$ with each sample in $J$ till a
    must-link is reached or all samples in $J$
    have been queried
  **if** $i$ *is must-linked to a sample from* $J_r$ *in* $J$
    **then**
    Move $i$ from $w$ to subcluster $J_r$
  **else**
    Move $i$ from $w$ to an empty subcluster,
      and add the new subcluster $\{i\}$ to $\mathcal{N}$
  **end if**
**end for**

**Algorithm 3:** Fast Transitive Inference

**Input:** State matrix $S$, new constraints $(s, t)$.
**for** $i$ *in* $(s, t)$ **do**
  Get $\mathcal{ML} = \{j | S[i, j] = 1\}$
  Get $\mathcal{CL} = \{j | S[i, j] = -1\}$
  Let $S[p, q] = 1$, for $p, q \in \mathcal{ML}$
  Let $S[p, q] = -1$, for $p \in \mathcal{ML}, q \in \mathcal{CL}$
**end for**

## 4.1 EXPERIMENTAL SETTING

**Datasets.** We sampled eight datasets from four representative real-world image sources to serve as our experimental data sources: `Market-1501` (Zheng et al., 2015), which comprises human body images from 1501 individuals.; `Humbi` (Yu et al., 2020), a large multiview image dataset focused on human expressions like faces; `Handwritten` (Dua et al., 2017), a collection containing 2000 samples of handwritten digits from '0' to '9'. (4) `MS1M` (Guo et al., 2016), a substantial benchmark dataset commonly used in face recognition tasks. To facilitate a meaningful comparison with state-of-the-art ACC methods, which are typically evaluated on datasets with fewer than 3000 samples, we have extracted four subsets from these image sources which are denoted as `MK20`, `MK100`, `Handwritten` and `Humbi-Face`. To assess the efficacy of our proposed APC framework on larger datasets, we have further selected four sizable subsets: `Humbi-Large`, `MS1M-10k`, `MS1M-100k` and `MK500`. The details of these datasets are shown in Table 1.

**Baselines.** We compare APC with two groups of work: the semi-supervised clustering methods that randomly select pairwise constraints (*Random-S* (Wauthier et al., 2012), *Random-P* (Basu et al., 2003)), and three representative ACC methods (*FFQS* (Basu et al., 2004), *URASC* (Xiong et al., 2016) and *NPU* (Xiong et al., 2013)). *Random-P* applies the PCKMeans (Basu et al., 2003) to update labels with randomly selected pairwise constraints. *Random-S* uses constrained spectral clustering (Wauthier et al., 2012) for label adjustment. *FFQS* uses the farthest-first scheme to acquire samples for better initialization. *NPU* is an ACC framework applicable to any semi-supervised clustering methods. *URASC* is an active spectral clustering method that iteratively selects samples that maximally reduce the uncertainty of the dataset.

**Implementation.** For the baseline methods, we maintain the same hyperparameter settings as reported in their original papers to ensure fairness in the comparison. For the initialization of APC, we set the number of neighbors to 50 when performing fast probabilistic clustering (Liu et al., 2022). As for the choice of hyperparameters, we set the epoch $T$ as 2 for all datasets. In addition, the setting of probability threshold $\mathbb{P}_l$ and compactness threshold $\tau$ are shown in Table 5 in Appendix C.1.

**Evaluation.** As discussed in Vinh et al. (2009), NMI can exhibit bias towards fine-grained clustering. Therefore, in addition to NMI, we employ the Adjusted Rand Index (ARI) (Hubert & Arabie, 1985) to evaluate the performance of APC and baseline ACC methods. Both NMI and ARI fall within the range of (0,1], with larger values indicating superior clustering performance. To further investigate whether APC effectively mitigates the category fission problem, we introduce two supplementary metrics: (1) the Fission Rate ($\Upsilon = \frac{k}{K}$), where $K$ is the number of underlying classes and

Table 1: This table presents details about the sampled datasets. Moreover, datasets in which each class contains an equal number of samples are categorized as balanced datasets.

|  | MK20 | MK100 | Handwritten | Humbi-Face | Humbi-Large | MS1M-10k | MS1M-100k | MK500 |
|---|---|---|---|---|---|---|---|---|
| $N$ | 351 | 1650 | 2000 | 2240 | 11200 | 10000 | 100000 | 9393 |
| $K$ | 20 | 100 | 10 | 40 | 200 | 146 | 1469 | 500 |
| balanced | ✗ | ✗ | ✓ | ✓ | ✓ | ✗ | ✗ | ✗ |

Figure 2: Performance comparison between APC and baselines on four datasets concerning the number of queries. More queries are invested in baselines to demonstrate their difference.

$k$ is the number of resulting clusters; and (2) the entropy of a partition $\Omega$, denoted as $\mathbb{H}(\Omega)$. When $\Upsilon$ approaches 1 and the entropy of the resulting clusters by APC approaches the entropy of real class partitions, we conclude that APC has effectively mitigated the category fission problem.

## 4.2 PERFORMANCE COMPARISON

We aim to evaluate the performance of APC ($\mathbb{P}_+$ in Eq. (5) is used) and baselines on multiple datasets with varying characteristics. Specifically, we examine how dataset size, the number of underlying classes, and class balance impact the effectiveness of these methods. To accomplish this, we conducted a comparative analysis of APC and five baselines on datasets including MK20, MK100, Handwritten and Humbi-Face, which collectively encompass the properties we intend to investigate. In Figure 2, we illustrate how the NMI and ARI results evolve with the investment of more human queries. Our observations are as follows:

- APC consistently outperforms all baseline methods, achieving NMI and ARI scores of 0.9 or higher, with absolute NMI and ARI gains of at least 0.08 and 0.3, respectively. Notably, this is achieved with far fewer queries than the dataset size $N$. However, only NPU matches this level of performance on MK20, but it requires six times as many queries.

Table 2: More experiment details about the implementation of APC on five datasets: (i) the time cost of APC (seconds); (ii) to what extent has APC alleviated the category fission problem.

| | Time (s) | $\Upsilon_{\mathrm{fpc}}$ | $\Upsilon_{\mathrm{apc}}$ | $\mathbb{H}_{\mathrm{real}}$ | $\mathbb{H}_{\mathrm{fpc}}$ | $\mathbb{H}_{\mathrm{apc}}$ |
|---|---|---|---|---|---|---|
| MK20 | 0.28 | 2.10 | 1.10 | 2.67 | 3.39 | 2.80 |
| MK100 | 5.18 | 1.76 | 0.94 | 4.28 | 4.68 | 4.69 |
| Handwritten | 1.88 | 10.50 | 3.10 | 2.30 | 3.88 | 2.52 |
| Humbi-Face | 2.46 | 3.22 | 1.10 | 3.69 | 4.53 | 3.71 |

Table 3: Ablation study on the query strategy of APC: (i) the ARI performance comparison of APC when different types of merge probability are used, under the same maximum query cost $Q_{\mathrm{max}}$; (ii) how the probability threshold, $\mathbb{P}_l$ influence the ARI performance of APC.

| | $\mathbb{P}$ | $\mathbb{P}_+$ | $\mathbb{P}_\times$ | $\mathbb{P}_c$ | $Q_{\mathrm{max}}$ | 0.3 | 0.4 | 0.5 | 0.6 | $Q_{\mathrm{max}}$ |
|---|---|---|---|---|---|---|---|---|---|---|
| MK20 | 0.807 | 0.957 | 0.940 | 0.953 | 60 | 0.916 | 0.836 | 0.836 | 0.834 | 30 |
| MK100 | 0.542 | 0.782 | 0.731 | 0.722 | 200 | 0.749 | 0.734 | 0.728 | 0.699 | 150 |
| Handwritten | 0.502 | 0.915 | 0.914 | 0.805 | 100 | 0.829 | 0.888 | 0.911 | 0.913 | 90 |
| Humbi-Face | 0.715 | 0.891 | 0.820 | 0.829 | 100 | 0.786 | 0.847 | 0.884 | 0.912 | 100 |
| Humbi-Large | 0.648 | 0.844 | 0.823 | 0.758 | 600 | 0.691 | 0.747 | 0.828 | 0.858 | 600 |

- All baselines perform well in two scenarios: when the dataset is small (MK20) or when the number of underlying classes is small (Handwritten). However, they struggle to provide robust improvements when $N$ is large and $K$ is slightly larger than 10 (Humbi-Face). They almost fail when $K$ is large, and the sample distribution is unbalanced across different classes (MK100). In contrast, APC consistently enhances clustering quality significantly with a limited number of queries, demonstrating its robustness across all three factors.

- APC possesses Monotonicity property, a result attributed to our Human Test, which ensures that APC adheres to the constraints outlined in Theorem 1. In contrast, all baseline methods occasionally degrade clustering performance when more queries are incorporated. This observation suggests that genuine supervisory information can sometimes be detrimental to clustering, as it may introduce conflicts (e.g., $(x_i, x_j)$ is cannot-linked, but their similarity to $x_k$ is larger than 0.9) in sample distribution and lead to contradictions within semi-supervised clustering, as discussed in Davidson et al. (2006).

**Category Fission.** For each dataset, we report the fission rate of the clustering results obtained using FPC (Liu et al., 2022) as $\Upsilon_{\mathrm{fpc}}$, along with the final fission rate after applying APC, denoted as $\Upsilon_{\mathrm{apc}}$. Additionally, we provide entropy values of the resulting clustering ($\mathbb{H}_{\mathrm{fpc}}$ and $\mathbb{H}_{\mathrm{apc}}$) and the ground truth clustering, denoted as $\mathbb{H}_{\mathrm{real}}$. The comprehensive results are tabulated in Table 3. Our observations show a significant reduction in the fission rate and entropy for all datasets when employing APC. This result reveals why APC is so effective: APC can accurately identify cluster pairs that should be merged, hence alleviating the category fission problem in FPC fast.

## 4.3 ABALATION STUDY

In this section, we conduct a detailed ablation study to show the influence of APC's query strategy on its performance across five datasets with varying sizes, spanning from 351 to 11,200 instances.

**Variants of Merge Probability.** We conduct an ablation study on APC, focusing on assessing how different merge probability types affect its performance. In addition to $\mathbb{P}$ in Eq. (2), $\mathbb{P}_\times$ in Eq. (3), and $\mathbb{P}_+$ in Eq. (4), we also explore the application of a classical cluster-based ACC query strategy, which utilizes the distance between centroid samples. To align with other types of merge probability, we map the distance to pairwise posterior probability, denoted as $\mathbb{P}_c$. We test these types of merge probability in five datasets and keep the recalled cluster pair numbers fixed for a fair comparison. As ARI is less biased than NMI and better depicts the impact of APC, we report their ARI performance under a predefined maximum query time threshold in Table 3, the related NMI results are shown in Table 6 in Appendix C.2. We observe that $\mathbb{P}_\times$ and $\mathbb{P}_+$, two variants of $\mathbb{P}$, are more accurate when recalling mergeable cluster pairs. This supports the rationale of only preserving the knn pairs in $\mathbb{P}_\times$ and $\mathbb{P}_+$. Additionally, we observe that $\mathbb{P}_c$ achieves comparable performance with $\mathbb{P}_+$ and $\mathbb{P}_\times$ only in the smallest dataset MK20. This underscores the superiority of our derived merge probability over the traditional centroid distance strategy, particularly in the context of larger datasets.

**Selection of Probability Threshold.** As $\mathbb{P}_+$ consistently outperforms other merge probabilities, we explore how to select an appropriate probability threshold. We apply APC to five datasets using

four different $\mathbb{P}_l$, and the results are in Table 3. We observe that as the size of the dataset increases, larger $\mathbb{P}_l$ will cause better performance for APC. Because setting a higher $\mathbb{P}_l$ leads to an increased probability of the query result being must-link, which will result in a higher merge success rate.

**Performance on Large datasets.** We further test APC on three large datasets `MS1M-10k`, `MS1M-100k` and `MK500`, and the result is in Table 4. We observe that APC could still significantly improve the clustering performance with a limited number of queries. Notably, APC takes less than 4000 queries to improve the ARI of `MS1M-100k` by 0.136 in less than 3 hours. Contrastingly, URASC and NPU exhibit limited capacity to significantly enhance the initial clustering performance of

Table 4: Tests on Large datasets.

|  | MS1M-10k | MS1M-100k | MK500 |
|---|---|---|---|
| Time cost | 59.45s | 2h47m | 56.48s |
| Query number | 621 | 3920 | 1061 |
| Initial NMI | 0.893 | 0.926 | 0.877 |
| Final NMI | 0.972 | 0.957 | 0.900 |
| Initial ARI | 0.730 | 0.742 | 0.450 |
| Final ARI | 0.947 | 0.878 | 0.591 |

`MS1M-100k`, even with an investment of 10k queries. Furthermore, their effectiveness diminishes notably when the initial clustering result is relatively good. For example, when the initial NMI performance of URASC on `MK100` is 0.64, it requires 5000 queries to improve it to 0.78. However, if the initial NMI performance of URASC reaches 0.76 on `MK100`, investing 5000 queries can even lead to a degradation in the NMI value. More detailed results about the performance of APC are shown in Figure 3 in Appendix C.3.

## 5 RELATED WORK

**Query Strategy in ACC.** Recent ACC methods have embraced a trend of incorporating sample uncertainty into their query strategies. These methods frequently utilize entropy to quantify uncertainty (Xiong et al., 2013; Abin, 2016; Xiong et al., 2016; Shi et al., 2020). A common task involves estimating the probability of a sample belonging to different clusters or neighborhoods. Additionally, alternative criteria such as maximum expected error reduction (Wang & Davidson, 2010) and maximum expected clustering change (Biswas & Jacobs, 2014) have been proposed to assess the stability of clustering results when perturbing the similarity values between two samples.

**Constraints in ACC.** When using the must-link and cannot-link constraints to perform constrained clustering, two aspects are usually taken into consideration: the transitive inference of constraints and whether the constraints can be violated. Transitive inference involves the derivation of additional constraints through an initial set of queried constraints. Notably, Lutz et al. (2021) have proposed an effective solution for implementing transitive inference within their pair-based active clustering methods. In terms of utilizing the constraints, ACC methods usually allow them to be broken, as they mostly adopt the classic soft Constrained Clustering algorithms like MPCKMeans (Basu et al., 2003) and ASC (Wauthier et al., 2012). In contrast, we adopt a relabeling strategy to update the cluster identities of samples, which does not violate the query result and is extremely time-saving compared to soft-constrained clustering methods.

**Estimation of Pairwise Probability.** Estimating the matching probability between samples is crucial in both probabilistic clustering (Lu & Leen, 2004) and some ACC methods (Biswas & Jacobs, 2014; Mai et al., 2013). Recently, Liu et al. (2022) have introduced using isotonic regression to learn a function that maps the distance or similarity between samples to the posterior pairwise probability. They have also derived the formula for aggregating pairwise probabilities in multi-view scenarios.

## 6 CONCLUSION

This paper studies Active Constrained Clustering (ACC) problems and aims to develop a cluster-based algorithm capable of handling large datasets. Based on a new theoretical result regarding the necessary condition of cluster fusion, we propose a new Active Probabilistic Clustering (APC) framework. Extensive experiments demonstrate the superiority of APC when compared to existing baseline methods, particularly when confronted with large datasets. Our work sheds light on the active clustering algorithms for large datasets. It would be interesting to explore the performance of APC or develop some new algorithms for larger datasets involving millions to billions of data.

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

# A PROOFS

## A.1 PROOF OF THEOREM 1

*Proof of Theorem 1.* Let $p$ and $q$ denote the sizes of $w_1$ and $w_2$, respectively. We further assume that the class index of these outliers is $i_j \in \{1, 2, \cdots, K\}$, where $j \in \{1, 2, \cdots, (1 - t_1)p + (1 - t_2)q\}$. For ease of presentation, for any class index $i \in \{1, 2, \cdots, K\}$, we use $s_i$ to denote the class size, i.e., $|c_i| = s_i$. Without loss generality, we assume that $q \geq p$ throughout this proof.

By the definition of mutual information, we have

$$\mathbb{I}(\Omega^*; C) = \mathbb{I}(\Omega; C) + \sum_{\tau=1}^{K} \mathbb{P}(w_{1,2} \cap c_\tau) \log \frac{\mathbb{P}(w_{1,2} \cap c_\tau)}{\mathbb{P}(w_{1,2})\mathbb{P}(c_\tau)} - \sum_{\tau=1}^{K} \mathbb{P}(w_1 \cap c_\tau) \log \frac{\mathbb{P}(w_1 \cap c_\tau)}{\mathbb{P}(w_1)\mathbb{P}(c_\tau)}$$

$$- \sum_{\tau=1}^{K} \mathbb{P}(w_2 \cap c_\tau) \log \frac{\mathbb{P}(w_2 \cap c_\tau)}{\mathbb{P}(w_2)\mathbb{P}(c_\tau)}$$

$$= \mathbb{I}(\Omega; C) + \underbrace{\sum_{\tau=1}^{K} \frac{|w_{1,2} \cap c_\tau|}{N} \log \frac{N \cdot |w_{1,2} \cap c_\tau|}{|w_{1,2}| \cdot |c_\tau|}}_{\text{(I)}} - \underbrace{\sum_{\tau=1}^{K} \frac{|w_1 \cap c_\tau|}{N} \log \frac{N \cdot |w_1 \cap c_\tau|}{|w_1| \cdot |c_\tau|}}_{\text{(II)}}$$

$$- \underbrace{\sum_{\tau=1}^{K} \frac{|w_2 \cap c_\tau|}{N} \log \frac{N \cdot |w_2 \cap c_\tau|}{|w_2| \cdot |c_\tau|}}_{\text{(III)}}. \tag{7}$$

Then we bound these three terms respectively. For Term (I), we have

$$\text{(I)} = \frac{|w_{1,2} \cap c_1|}{N} \log \frac{N \cdot |w_{1,2} \cap c_1|}{|w_{1,2}| \cdot |c_1|} + \sum_{\tau=2}^{K} \frac{|w_{1,2} \cap c_\tau|}{N} \log \frac{N \cdot |w_{1,2} \cap c_\tau|}{|w_{1,2}| \cdot |c_\tau|}$$

$$= \frac{t_1 p + t_2 q}{N} \log \frac{N(t_1 p + t_2 q)}{(p + q)s_1} + \sum_{j=1}^{(1-t_1)p+(1-t_2)q} \frac{1}{N} \log \frac{N \cdot |w_{1,2} \cap c_{i_j}|}{(p + q)s_{i_j}}. \tag{8}$$

Similarly, we have

$$\text{(II)} = \frac{|w_1 \cap c_1|}{N} \log \frac{N \cdot |w_1 \cap c_1|}{|w_1| \cdot |c_1|} + \sum_{\tau=2}^{K} \frac{|w_1 \cap c_\tau|}{N} \log \frac{N \cdot |w_1 \cap c_\tau|}{|w_1| \cdot |c_\tau|}$$

$$= \frac{t_1 p}{N} \log \frac{N t_1}{s_1} + \sum_{j=1}^{(1-t_1)p} \frac{1}{N} \log \frac{N \cdot |w_1 \cap c_{i_j}|}{p s_{i_j}}, \tag{9}$$

and

$$\text{(III)} = \frac{|w_2 \cap c_1|}{N} \log \frac{N \cdot |w_2 \cap c_1|}{|w_2| \cdot |c_1|} + \sum_{\tau=2}^{K} \frac{|w_2 \cap c_\tau|}{N} \log \frac{N \cdot |w_2 \cap c_\tau|}{|w_2| \cdot |c_\tau|}$$

$$= \frac{t_2 q}{N} \log \frac{N t_2}{s_2} + \sum_{j=1}^{(1-t_2)q} \frac{1}{N} \log \frac{N \cdot |w_2 \cap c_{i_j}|}{q s_{i_j}}. \tag{10}$$

Plugging Eq. (8), Eq. (9), and Eq. (10) into Eq. (7), together with the fact that

$$|w_{1,2} \cap c_{i_j}| \geq \max\{|w_1 \cap c_{i_j}|, |w_2 \cap c_{i_j}|\}, \quad \forall j \in \{1, 2, \cdots, (1 - t_1)p + (1 - t_2)q\},$$

we obtain that

$$\mathbb{I}(\Omega^*; C) \geq \mathbb{I}(\Omega; C) + \underbrace{\frac{t_1 p}{N} \log \frac{t_1 p + t_2 q}{t_1(p + q)} + \frac{t_2 q}{N} \log \frac{t_1 p + t_2 q}{t_2(p + q)}}_{\text{(IV)}} \tag{11}$$

$$- \underbrace{\left( \frac{(1 - t_1)p + (1 - t_2)q}{N} \log(p + q) - \frac{(1 - t_1)p}{N} \log p - \frac{(1 - t_2)q}{N} \log q \right)}_{\text{(V)}}. \tag{12}$$

**Term (IV) in Eq.** (11). For Term (IV) in Eq. (11), by the Taylor expansion

$$\log(1+x) = \sum_{u=1}^{\infty} \frac{(-1)^{u-1}}{u} \cdot x^u,$$

we have

$$
\begin{aligned}
\text{(IV)} &= \frac{t_1 p}{N} \sum_{u=1}^{\infty} \frac{(-1)^{u-1}}{u} \cdot \left(\frac{(t_2 - t_1)q}{t_1(p+q)}\right)^u + \frac{t_2 q}{N} \sum_{u=1}^{\infty} \frac{(-1)^{u-1}}{u} \cdot \left[\frac{(t_1 - t_2)p}{t_2(p+q)}\right]^u \\
&= \sum_{v=1}^{\infty} \frac{1}{2v-1} \cdot \left[\frac{t_1 p}{N} \cdot \left(\frac{(t_2 - t_1)q}{t_1(p+q)}\right)^{2v-1} + \frac{t_2 q}{N}\left(\frac{(t_1 - t_2)p}{t_2(p+q)}\right)^{2v-1}\right] \\
&\quad - \sum_{v=1}^{\infty} \frac{1}{2v} \cdot \left[\frac{t_1 p}{N} \cdot \left(\frac{(t_2 - t_1)q}{t_1(p+q)}\right)^{2v} + \frac{t_2 q}{N}\left(\frac{(t_1 - t_2)p}{t_2(p+q)}\right)^{2v}\right].
\end{aligned}
\tag{13}
$$

For ease of presentation, we denote $m = q/p \geq 1$. Then for any $v \geq 1$, we have

$$
\begin{aligned}
\sum_{v=1}^{\infty} \frac{1}{2v} &\cdot \left[\frac{t_1 p}{N} \cdot \left(\frac{(t_2 - t_1)q}{t_1(p+q)}\right)^{2v} + \frac{t_2 q}{N}\left(\frac{(t_1 - t_2)p}{t_2(p+q)}\right)^{2v}\right] \\
&= \sum_{v=1}^{\infty} \frac{pq(t_1 - t_2)^{2v}}{2vN(p+q)^{2v}} \cdot \left[\frac{q^{2v-1}}{t_1^{2v-1}} + \frac{p^{2v-1}}{t_2^{2v-1}}\right] \\
&\leq \sum_{v=1}^{\infty} \frac{mp^2(3/10)^{2v}}{2vNp^{2v}(1+m)^{2v}} \cdot (1 + m^{2v-1}) \cdot \left(\frac{10p}{7}\right)^{2v-1} \\
&\leq \frac{3p}{20N} \sum_{v=1}^{\infty} \frac{1}{v} \cdot \left(\frac{3}{7}\right)^{2v-1} \leq \frac{0.0716p}{N},
\end{aligned}
\tag{14}
$$

where the first inequality uses $m = q/p$ and the assumption that $t_1, t_2 \in [0.7, 1]$, the second inequality follows the fact that $m(1 + m^{2v-1}) \leq (1+m)^{2v}$, and the last inequality follows that

$$\sum_{v=1}^{\infty} \frac{1}{v} \cdot \left(\frac{3}{7}\right)^{2v-1} \leq \frac{3}{7} + \sum_{v=2}^{\infty} \frac{1}{2} \cdot \left(\frac{3}{7}\right)^{2v-1} = \frac{267}{560}$$

and simple calculations.

On the other hand, for any $v \geq 1$, we have

$$
\begin{aligned}
\sum_{v=1}^{\infty} \frac{1}{2v-1} &\cdot \left[\frac{t_1 p}{N} \cdot \left(\frac{(t_2 - t_1)q}{t_1(p+q)}\right)^{2v-1} + \frac{t_2 q}{N}\left(\frac{(t_1 - t_2)p}{t_2(p+q)}\right)^{2v-1}\right] \\
&= \sum_{v=1}^{\infty} \frac{pq(t_1 - t_2)^{2v-1}}{(2v-1)N(p+q)^{2v-1}} \cdot \left[\frac{p^{2v-2}}{t_2^{2v-2}} - \frac{q^{2v-2}}{t_1^{2v-2}}\right] \\
&= \sum_{v=1}^{\infty} \frac{pq(t_1 - t_2)^{2v+1}}{(2v+1)N(p+q)^{2v+1}} \cdot \left[\frac{p^{2v}}{t_2^{2v}} - \frac{q^{2v}}{t_1^{2v}}\right].
\end{aligned}
\tag{15}
$$

Furthermore, we have

$$
\begin{aligned}
\sum_{v=1}^{\infty} \frac{pq(t_1 - t_2)^{2v+1}}{(2v+1)(p+q)^{2v+1}} \cdot \left[\frac{p^{2v}}{t_2^{2v}} - \frac{q^{2v}}{t_1^{2v}}\right] &\geq - \sum_{v=1}^{\infty} \frac{pq|t_1 - t_2|^{2v+1}}{(2v+1)N(p+q)^{2v+1}} \cdot \left[\frac{p^{2v}}{t_2^{2v}} + \frac{q^{2v}}{t_1^{2v}}\right] \\
&\geq -\frac{p}{N} \sum_{v=1}^{\infty} \frac{1}{2v+1} \cdot \left(\frac{3}{10}\right)^{2v+1} \\
&\geq -\frac{0.0096p}{N},
\end{aligned}
\tag{16}
$$

where the second inequality uses the facts that $(p+q)^{2v+1} \geq q^{2v+1} + qp^{2v}$ and $t_1, t_2 \in [0.7, 1]$, and the last inequality follows that

$$\sum_{v=1}^{\infty} \frac{1}{2v+1}\left(\frac{3}{10}\right)^{2v+1} \leq \frac{9}{1000} + \frac{1}{5}\sum_{v=2}^{\infty}\left(\frac{3}{10}\right)^{2v+1} = \frac{9}{1000} + \frac{243}{455000} < 0.0096.$$

Combining Eq. (13), Eq. (14), Eq. (15), and Eq. (16), we obtain that

$$\text{(IV)} \geq -\frac{0.0812p}{N}.\tag{17}$$

**Term (V) in Eq. (12).** For Term (V) in Eq. (12), we have

$$(\mathrm{V}) = \frac{(1-t_1)p + (1-t_2)q}{N} \log(p+q) - \frac{(1-t_1)p}{N} \log p - \frac{(1-t_2)q}{N} \log q$$

$$= \frac{(1-t_1)p}{N} \log \frac{p+q}{p} + \frac{(1-t_2)q}{N} \log \frac{p+q}{q}. \tag{18}$$

Furthermore, we have

$$\frac{(1-t_1)p}{N} \log \frac{p+q}{p} + \frac{(1-t_2)q}{N} \log \frac{p+q}{q} \tag{19}$$

$$\leq \frac{(1-\min\{t_1,t_2\})p}{N} \log \frac{p+q}{p} + \frac{(1-\min\{t_1,t_2\})q}{N} \log \frac{p+q}{q}$$

$$= \frac{(1-\min\{t_1,t_2\})}{N} \cdot \Big[ p \log \frac{p+q}{p} + q \log \frac{p+q}{q} \Big]. \tag{20}$$

Let

$$\Delta h = \frac{1}{N} \cdot \Big[ p \log \frac{p+q}{p} + q \log \frac{p+q}{q} \Big]. \tag{21}$$

Combining Eq. (18), Eq. (19), and Eq. (21), we obtain

$$(\mathrm{V}) \leq (1-\min\{t_1,t_2\}) \cdot \Delta h. \tag{22}$$

**Putting Together.** Plugging Eq. (17) and Eq. (22) into Eq. (11) and Eq. (12), we have

$$\mathbb{I}(\Omega^*; C) \geq \mathbb{I}(\Omega; C) - \frac{0.0812p}{N} - (1-\min\{t_1,t_2\}) \cdot \Delta h. \tag{23}$$

Recall that the $\Delta h$ defined in Eq. (21) takes the form

$$\Delta h = \frac{1}{N} \cdot \Big[ p \log \frac{p+q}{p} + q \log \frac{p+q}{q} \Big]$$

$$= \frac{p}{N} \cdot \Big( \log(1+m) + m \log \Big(1 + \frac{1}{m}\Big) \Big)$$

$$\geq 2 \log 2 \cdot \frac{p}{N}, \tag{24}$$

where the second equality uses $m = q/p$, the last inequality uses $m \geq 1$. Putting Eq. (23) and Eq. (24) together, we have

$$\mathbb{I}(\Omega^*; C) \geq \mathbb{I}(\Omega; C) - (1.0586 - \min\{t_1,t_2\}) \cdot \Delta h. \tag{25}$$

Then, we calculate the entropy after fusion.

$$\mathbb{H}(\Omega^*) = \mathbb{H}(\Omega) - \frac{p+q}{N} \log \frac{p+q}{N} + \frac{p}{N} \log \frac{p}{N} + \frac{q}{N} \log \frac{q}{N}$$

$$= \mathbb{H}(\Omega) - \frac{p+q}{N} \log(p+q) + \frac{p}{N} \log p + \frac{q}{N} \log q$$

$$= \mathbb{H}(\Omega) - \Delta h \tag{26}$$

Recall that

$$n_1 = \frac{2\mathbb{I}(\Omega; C)}{\mathbb{H}(\Omega) + \mathbb{H}(C)}, \quad n_2 = \frac{2\mathbb{I}(\Omega^*; C)}{\mathbb{H}(\Omega^*) + \mathbb{H}(C)}.$$

By Eq. (25) and Eq. (26), we know that the sufficient condition of $n_2 \geq n_1$ is

$$\frac{2[\mathbb{I}(\Omega; C) - (1.0586 - \min\{t_1,t_2\}) \cdot \Delta h]}{\mathbb{H}(\Omega) + \mathbb{H}(C) - \Delta h} \geq \frac{2\mathbb{I}(\Omega; C)}{\mathbb{H}(\Omega) + \mathbb{H}(C)},$$

which is equivalent to

$$n_1 = \frac{2\mathbb{I}(\Omega; C)}{\mathbb{H}(\Omega) + \mathbb{H}(C)} \geq 2 \cdot (1.0586 - \min\{t_1,t_2\}),$$

which concludes the proof of Theorem 1. $\qquad\square$

## A.2 DERIVATION OF EQ. (2)

*Derivation of Eq.* (2). With the assumption of mutual independence among individual events $e_{st}$, we can articulate the joint probability density of a clustering $\pi = [z_1, z_2, \cdots, z_m]$ for $m$ samples as $\mathbb{P}(\pi) = \frac{1}{\alpha} \prod_{s,t \in [1,2,\cdots,m]} \mathbb{P}(e_{st} = 1)^{I(z_s = z_t)} \times \mathbb{P}(e_{st} = 0)^{I(z_s \neq z_t)}$, where $I(\cdot)$ is the indicator function, and $\alpha$ is the normalization factor (Liu et al., 2022).

Under the condition of $w_i$ and $w_j$ (i.e., $\forall (s,t) \in w_i$ or $w_j, e_{st} = 1$), the events of $w_i = w_j$ (i.e., $\forall s \in w_i, \forall t \in w_j, e_{st} = 1$) and $w_i \neq w_j$ (i.e., $\forall s \in w_i, \forall t \in w_j, e_{st} = 0$) are mutually exclusive. Therefore, by the formula of conditional probability, we can obtain:

$$
\begin{aligned}
\mathbb{P}(w_i = w_j | w_i, w_j) &= \frac{\mathbb{P}(w_i = w_j, w_i, w_j)}{\mathbb{P}(w_i, w_j)} \\
&= \frac{\mathbb{P}(w_i = w_j, w_i, w_j)}{\mathbb{P}(w_i = w_j, w_i, w_j) + \mathbb{P}(w_i \neq w_j, w_i, w_j)} \\
&= \frac{\frac{1}{A} \prod_{s \in w_i, t \in w_j} \mathbb{P}(e_{st} = 1) \prod_{s, t \in w_i, s, t \in w_j} \mathbb{P}(e_{st} = 1)}{\frac{1}{A} [\prod_{s \in w_i, t \in w_j} \mathbb{P}(e_{st} = 1) + \prod_{s \in w_i, t \in w_j} \mathbb{P}(e_{st} = 0)] \prod_{s, t \in w_i, s, t \in w_j} \mathbb{P}(e_{st} = 1)} \\
&= \frac{\prod_{s \in w_i, t \in w_j} \mathbb{P}(e_{st} = 1)}{\prod_{s \in w_i, t \in w_j} \mathbb{P}(e_{st} = 1) + \prod_{s \in w_i, t \in w_j} \mathbb{P}(e_{st} = 0)},
\end{aligned}
$$

where $A$ is the normalization factor of the joint probability density for samples in $\{w_i, w_j\}$. $\qquad \square$

## A.3 JUSTIFICATION OF APPROXIMATION

**Regarding the Mutual Information.** If the purity of $w_1$ and $w_2$ is 1, and they belong to the same class $c_\tau$, then we have

$$
\mathbb{P}(w_1 \cap c_\tau) = \mathbb{P}(w_1), \quad \mathbb{P}(w_2 \cap c_\tau) = \mathbb{P}(w_2), \quad \mathbb{P}(w_{1,2} \cap c_\tau) = \mathbb{P}(w_{1,2}) = \mathbb{P}(w_1) + \mathbb{P}(w_2).
$$

Hence, and we have $\mathbb{I}(\Omega^*; C) = \mathbb{I}(\Omega; C)$. This is because

$$
\begin{aligned}
\mathbb{I}(\Omega^*; C) &= \mathbb{I}(\Omega; C) + \mathbb{P}(w_{1,2} \cap c_\tau) \log \frac{\mathbb{P}(w_{1,2} \cap c_\tau)}{\mathbb{P}(w_{1,2}) \mathbb{P}(c_\tau)} \\
&\quad - \mathbb{P}(w_1 \cap c_\tau) \log \frac{\mathbb{P}(w_1 \cap c_\tau)}{\mathbb{P}(w_1) \mathbb{P}(c_\tau)} - \mathbb{P}(w_2 \cap c_\tau) \log \frac{\mathbb{P}(w_2 \cap c_\tau)}{\mathbb{P}(w_2) \mathbb{P}(c_\tau)} \\
&= \mathbb{I}(\Omega; C) + \mathbb{P}(w_{1,2}) \log \frac{1}{\mathbb{P}(c_\tau)} - \mathbb{P}(w_1) \log \frac{1}{\mathbb{P}(c_\tau)} - \mathbb{P}(w_2) \log \frac{1}{\mathbb{P}(c_\tau)} \\
&= \mathbb{I}(\Omega; C).
\end{aligned}
$$

**Regarding the Approximation of Entropy.** If $(p + q) \ll N$, then

$$
\begin{aligned}
\Delta h &= \frac{p+q}{N} \log(p+q) - \frac{p}{N} \log p - \frac{q}{N} \log q \\
&< \frac{p+q}{N} \log(p+q) \\
&< \frac{p+q}{N} \log \frac{N}{p+q}.
\end{aligned}
$$

Note that $\mathbb{H}(\Omega) = \sum_s \frac{s}{N} \log \frac{N}{s}$, where $s$ is the cluster size like $p$ and $q$. Together with $p + q \ll N$, we have

$$
\Delta h \ll \mathbb{H}(\Omega) < \mathbb{H}(\Omega) + \mathbb{H}(C).
$$

## B ALGORITHM DETAILS OF APC

### B.1 PAIRWISE PROBABILITY ESTIMATION.

Following the setup in Liu et al. (2022), we train an isotonic regressor to estimate the pairwise posterior probability $\mathbb{P}(e_{ij} = 1 | d_{ij})$, where $d_{ij}$ is the Euclidean distance between samples $i$ and $j$. The estimation consists of three steps:

(1) As the real label is unavailable, we utilize the K-means clustering to generate the pseudo label for the samples.

(2) Then, we use the $k$-nearest-neighbors of each sample to generate sample pairs. We label these pairs as 0 or 1 using the pseudo labels, to indicate whether the two samples belong to the same class.

(3) Finally, we use isotonic regression to learn a function that maps the Euclidean distance between two samples to the probability that they belong to the same class.

Liu et al. (2022) also proposed Graph-context-aware refinement to enhance the quality of the posterior probability, but it is not an essential component of APC. Therefore, we did not include it in our experiments. However, incorporating them would further enhance the performance of APC, as they can improve the quality of the estimated merging probability between cluster pairs.

### B.2 IMPLEMENTATION OF PURITY TEST

To estimate the compactness of a cluster, we need to calculate $\sum_{j \in w(i)} \mathbb{P}(e_{ij} = 1)$ for every sample $i$ in w. In practice, we sort all samples in $w$ according to their pairwise probability with $i$ (i.e., $\mathbb{P}(e_{ij} = 1)$) in descending order, and put the samples whose rank falls within the range between $\lfloor \frac{|w|}{2} \rfloor$ and $\lfloor \frac{|w|}{2} + \sqrt{|w|} \rfloor$ into the set $w(i)$. After we get $w(i)$ for all samples in $w$, we calculate the compactness by $\frac{\sum_{i \in w, j \in w(i)} \mathbb{P}(e_{ij}=1)}{\sum_{i \in w} |w(i)|}$.

In the event that the compactness criterion is not met, we involve humans to judge the purity of the harsh cluster. Within a cluster, our belief is that samples usually exhibit a greater pairwise probability when compared to samples from the same class rather than samples from a different class. Hence, we select a sample pair from this cluster for the purity test as follows: (i) we first select the central sample of $w$, which is defined as $s_1$. (ii) next, we rank all samples in $w$ according to their pairwise probability with the $s_1$ in descending order. Then, we randomly select a sample whose rank falls within the range of $[\lfloor 0.5 * |w| \rfloor, \lfloor 0.7 * |w| \rfloor]$, and denote the sample as $s_2$. After selecting $(s_1, s_2)$, we let the humans judge if they belong to the same class. We claim that the cluster $w$ passes the Purity Test if the query result for this sample pair is a must-link.

### B.3 FAST TRANSITIVE INFERENCE

FTI involves expanding the set of constraints based on the information within the original set. For example, if $(x_1, x_2)$ and $(x_2, x_3)$ are must-link constraints, and $(x_1, x_4)$ is a cannot-link constraint, it implies that $(x_1, x_3)$ must be a must-link constraint, while $(x_2, x_4)$ and $(x_3, x_4)$ must be cannot-link constraints.

To facilitate the process, we present an efficient method, Fast Transitive Inference (FTI), which is designed to discover the transitive closure for the constraint set in APC. The implementation is shown in Algorithm 3.

The performance of FTI is guaranteed by Theorem 2.

**Theorem 2** (Completeness of FTI). *By executing the FTI algorithm every time a new human query is made, we can always get the latest transitive closure.*

*Proof of Theorem 2.* Suppose the must-link sample sets with samples $i$ and $j$ are denoted as $G_i$ and $G_j$, respectively. And the sample sets that are cannot-link with $i$ and $j$ are denoted as $g_i$ and $g_j$. When the constraints between $i$ and $j$ are queried, only the constraints of sample pairs within $\{G_i, G_j, g_i, g_j\}$ may change, as no must-link constraints are built between them and the rest samples. We discuss the two cases where $i$ and $j$ are must-linked or cannot-linked:

1 $(i, j)$ is must-linked. First, FTI updates the constraints for sample pairs related to $i$, then $G_i' = G_i \cup \{j\}$, $G_j' = G_j \cup G_i$, and $g_j' = g_j \cup g_i$. The constraints between $i$ and $G_j, g_j$ have not yet been updated. Then, FTI updates the constraints for sample pairs related to $j$, then we have $G_i' = G_i \cup G_j$ and $g_i' = g_i \cup g_j$. And this means all ml constraints between $G_i$ and $G_j$, and cl constraints between $\{G_i, G_j\}$ and $\{g_i, g_j\}$ are updated and stored in the state matrix $S$.

2 $(i, j)$ is cannot-linked. First, FTI updates the constraints for sample pairs related to $i$, then $g_i' = g_i \cup \{j\}$ and $g_j' = g_j \cup G_i$. Then FTI updates the constraints for sample pairs related to $j$, then we have $g_i' = g_i \cup G_j$. And this means all cl constraints between $G_i$ and $G_j$ are updated and stored in $S$.

Combining these two scenarios, we finish the proof of Theorem 2 □

## C  MORE EXPERIMENTAL DETAILS

### C.1  SELECTION OF PROBABILITY THRESHOLD.

We recommend selecting the probability threshold $\mathbb{P}_l$ from $\{0.7, 0.6, 0.5, 0.4, 0.3, 0.2, 0.1\}$. The merging probability threshold should be chosen such that the number of recalled cluster pairs is similar to the number of resulting clusters by FPC. Setting a higher threshold value increases the likelihood of cluster pairs belonging to the same class. This can enhance the efficiency of APC as shown in Table 3. Conversely, setting a lower threshold results in more recalled clusters, ultimately leading to NMI performance approaching 1 with additional human queries.

For the actual implementation of APC, we report our choice of hyperparameter in Table 5.

Table 5: This table presents details about the sampled datasets. Moreover, datasets in which each class contains an equal number of samples are categorized as balanced datasets.

|  | MK20 | MK100 | Handwritten | Humbi-Face | MK500 | MS1M-10k | MS1M-100k | Humbi-Large |
|---|---|---|---|---|---|---|---|---|
| $\mathbb{P}_l$ | 0.2 | 0.3 | 0.5 | 0.5 | 0.6 | 0.6 | 0.6 | 0.6 |
| $\tau$ | 0.8 | 0.8 | 0.7 | 0.7 | 0.85 | 0.7 | 0.7 | 0.7 |

### C.2  MORE ABLATION RESULTS

We report the NMI performance of the ablation study on APC in Table 6. We observe a similar trend as the ARI performance in Table 3: (i) $\mathbb{P}_+$ shows consistent advantage over other merge probability; (ii) Setting a larger probability threshold will improve the convergence speed of APC: reach a high level performance with fewer human queries.

Table 6: Ablation study about the query strategy of APC: (i) the NMI performance comparison of APC when different merge probability are used under the same max query limit, $Q_{\max}$; (ii) how the probability threshold, $\mathbb{P}_l$ affects the NMI performance of APC under the same max query cost $Q_{\max}$.

|  | NMI | | | | | NMI | | | | |
|---|---|---|---|---|---|---|---|---|---|---|
|  | $\mathbb{P}$ | $\mathbb{P}_+$ | $\mathbb{P}_\times$ | $\mathbb{P}_c$ | $Q_{\max}$ | 0.3 | 0.4 | 0.5 | 0.6 | $Q_{\max}$ |
| MK20 | 0.917 | 0.955 | 0.948 | 0.955 | 60 | 0.93 | 0.917 | 0.917 | 0.918 | 30 |
| MK100 | 0.862 | 0.905 | 0.899 | 0.896 | 200 | 0.895 | 0.895 | 0.897 | 0.891 | 150 |
| Handwritten | 0.746 | 0.918 | 0.916 | 0.867 | 100 | 0.872 | 0.896 | 0.911 | 0.913 | 90 |
| Humbi-Face | 0.903 | 0.958 | 0.937 | 0.937 | 100 | 0.918 | 0.938 | 0.954 | 0.965 | 100 |
| Humbi-Large | 0.913 | 0.953 | 0.944 | 0.935 | 600 | 0.916 | 0.926 | 0.946 | 0.956 | 600 |

### C.3  MORE RESULTS ON LARGE DATASETS

We show how APC enhances the clustering quality on large datasets in Figure 3.

### C.4  COMPLEXITY ANALYSIS

The major computational cost lies in calculating the merging probability and the expected NMI gain between all cluster pairs, with a computational complexity of $O(k^2)$, where $k$ is the number of resulting clusters from FPC (excluding outliers). Overall, the computational cost of APC is $O(T \cdot k^2)$, where $T$ is the number of epochs and is set to 2 in our experiments.

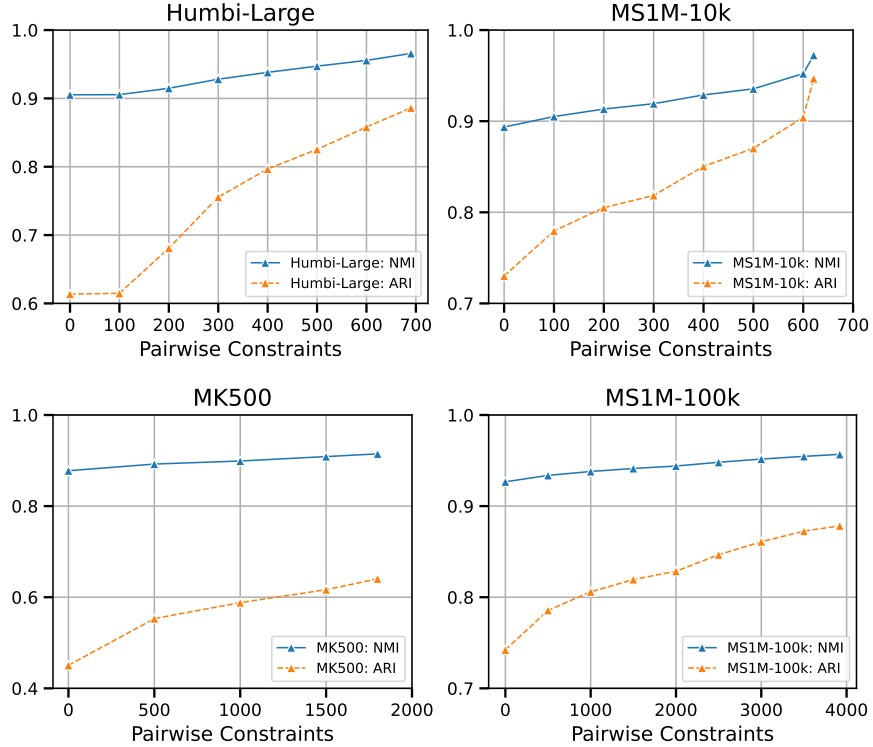

Figure 3: The NMI and ARI performance of APC on four large datasets.

## C.5 COMPUTING RESOURCES

We utilize a [GeForce RTX 3090 Ti] for feature extraction using DNN models. For the implementation of baseline methods and APC, we perform the experiments on a machine equipped with an Intel(R) Xeon(R) Platinum 8163 CPU @ 2.50GHz.

