# OpenReview forum: "Active Probabilistic Clustering"
_ICLR.cc/2024/Conference — ICLR 2024 Conference Withdrawn Submission_

### Official Review · Reviewer_TE49 · 2023-10-27

**Soundness:** 1 poor
**Presentation:** 1 poor
**Contribution:** 2 fair
**Rating:** 3
**Confidence:** 4

**Summary:**

This paper studies active constrained clustering, a semi-supervised learning problem in which the clustering method can query pairs of samples $(x_i, x_j)$ to learn if their underlying class labels $y_i, y_j$ are the same or different.  The goal seems to be to reduce the number of queries needed in order to learn the ground truth labels.

After developing a merging criterion based on normalized mutual information, a new active clustering method is proposed which the authors call active probabilistic clustering.  The idea is to construct appropriate sampling probabilities for pairs of clusters and to sample several pairs of clusters and query representatives from each cluster in order to make merging/splitting decisions.  Many experiments are carried out for this method, showing improvements to normalized mutual information and adjusted rand index using fewer queries than baseline methods.

**Strengths:**

Aside from the mathematics (see below), the writing and presentation is clear and seems to be well motivated.  The proposed method performs well in the experiments that were considered.

**Weaknesses:**

The problem formulation isn't clear at all aside from "the clustering method can query pairs of samples ...".  The actual objective and performance metrics are not clearly described or motivated in Sections 1 or 2.  I think it would help the reader to develop the objective for the problem more formally and contextualize prior work within this objective.

Additionally, without more care towards formalizing the problem setting and any mathematical models, I believe there are serious issues with how this paper is written.  In particular more care is needed to define the probability space and random variables that are being worked with to clarify the equations.  For example, Equation (2), seems to be wrong as written.  The LHS of this equation, ${\mathbb P}(w_i = w_j \mid w_i,w_j)$  has to either be 1 or 0, while the RHS does not.  The reason the LHS should be 1 or 0 is that once we've conditioned on the two random variables $w_i, w_j$, they are deterministically either the same or different.  If something else is meant, then this has to be clearly defined and communicated.  Similar issues hold for equations (3) and (4) as well.

**Questions:**

- Please clarify and motivate the objective (metrics) for the problem up front.
- Please clarify the terminology.  What is meant by splitting and clustering (e.g., in Theorem 2)?  The notations for these seem similar but I can't tell if splitting is just being used as a synonym for clustering or not.
- Please clarify what is meant by equations (2), (3), and (4).  What is the probability space, random variables, etc.?
- Please clarify the notation used.  What is meant by $w_j$?  Properly define entropy and mutual information, etc.

---

> ### Author Response · Authors · 2023-11-19
>
> Thank you for your and the other reviewers' valuable feedback. Taking into consideration the concerns raised regarding terminology, problem definitions, and algorithmic details, we have diligently revised the corresponding sections and added a flowchart illustrating how APC operates. We hope our response and the updated manuscript address your concerns and related issues about the paper. We answer your questions as follows:
>
> <Weakness 1, Question 1> *The problem formulation is not clearly described. Please clarify and motivate the objective (metrics) for the problem.*
>
> Thanks for your advice. Firstly, we study the Active Constrained Clustering (ACC) problem, which is formally defined in Section 2 Definition of ACC. ACC belongs to the scope of semi-supervised clustering, and it utilizes the actively queried pairwise constraints as supervisory information to strengthen the original unsupervised clustering result.
>
> Secondly, the objective of APC is to maximize the improvement of Normalized Mutual Information (NMI) with as few human queries as possible. This objective is motivated by the observed challenges in previous ACC works, specifically the "Accumulation Plane" problem encountered with complex datasets. In the revised version, we provide a detailed and clear explanation of our motivation for selecting this objective and using the NMI metric. This information is presented in Section 2.
>
>
> <Weakness 2, Question 3> *Considering that $w_i$ and $w_j$ are two random variables, the LHS of equation (2) has to be either 1 or 0, while the RHS of (2) does not. The same for equation (3) and (4). Please Clarify the probability space and random variables.*
>
> Thank you for your feedback. It's crucial to note that in Equation (2), $w_i$ and $w_j$ are not random variables. For the conditional merge probability expressions $P(w_i=w_j|w_i,w_j)$, the condition $w_i$ implies that the samples within this cluster share a common class, and the same interpretation applies to the condition $w_j$. $w_i=w_j$ signifies that the classes of the two clusters are identical. Note that the relationship between the classes of $w_i$ and $w_j$ remains unknown unless observed by human annotators. Therefore, we derive equations (2), (3) and (4) to estimate the probability of the event $w_i=w_j$ occurring.
>
> Common clustering algorithms are based on either the distance or the similarity between two samples. Similarly, the basic event (i.e., random variable) in probabilistic clustering is whether two samples belong to the same class or not: $e_{ij}\in \{1,0\}$, and its probability is defined as $P(e_{ij}=1)\in [0,1]$. More complex events like $(w_i=w_j|w_i,w_j)$ are the combinations of these basic events, and their probability is derived based on $P(e_{ij}=1)$.
>
> We have refined the explanation of these equations and added the discussion about probability space and random variables to minimize potential confusion in the revision.
>
> <Question 2> *Clarify the terminology of "splitting" and "clustering" in Theorem 2.*
>
> Thanks for your detailed review. In Theorem 2, the terms "clustering" and "splitting" convey the same concept. To eliminate potential confusion associated with the term "splitting,", we have changed the "splitting" to "clustering" in the Theorem in the revision.
>
> <Question 4> *Clarify the notation of $w_j$, entropy and mutual information.*
>
> $w_j$ first appears in Theorem 2, where it denotes a cluster that contains some samples. In Equation (2), $w_j$ as a condition means the event that all samples in this cluster have the same class. We have added the definition of entropy and mutual information in Definition 2.2 in the revision.

---

> > ### Comment · Reviewer_TE49 · 2023-11-21
> >
> > Thank you for your detailed reply.  I would like to focus on Question 3.  I can see better now what you are trying to communicate, but I still have some doubts:
> >
> > 1. The usage of equality is very non-standard.  In almost all of mathematics, we use "x = y" to refer to the statement that x and y are the exact same object.  The event you are trying to describe with $w_i = w_j | w_i, w_j$ is more like $\forall s \in w_i \ \forall t \in w_j, (e_{st}=1)$.  Similarly, when $w_i$ is written inside of a probability, it seems to mean $\forall s,t \in w_i, (e_{st} =1)$.  I would recommend introducing notation to clarify this as otherwise the equations appear misleading.
> >
> > 2. You can only base the probability of this joint event off of the probability of the marginal events by making assumptions (e.g., independence, which seems to be what is going on here from Appendix A.2), otherwise the calculation should be more complicated.   Please clarify and justify any assumptions you are making concerning probabilities.

---

> > > ### Author Response · Authors · 2023-11-22
> > >
> > > Thanks for your detailed review and feedback.
> > > - For the first question. We agree with what you recommend, and we have added the notation in Section 3.1 to avoid possible misunderstandings in the latest revision.
> > > - For the second question, both FPC and our work indeed assume the independence among different $e_{st}$. We have clarified the assumption in Section 3.1 and the proof in Appendix A.2 in the revision.

---

> > > > ### Comment · Reviewer_TE49 · 2023-11-22
> > > >
> > > > Thank you for the response and the detailed revisions.  I am inclined to agree with other reviewers that this paper should go through another round of revisions and reviewing.
> > > >
> > > > In particular, the presentation of the problem formulation is still shaky to me as I don't see how probabilities arise naturally.  Specifically, Definition 2.1 doesn't seem to suggest any probabilistic modeling.  Definition 2.2 is creating distributions over clusters artificially by just placing (size of cluster)/(number of points) mass at each cluster.  NMI then just seems to be giving a measure of how much two different clusterings overlap with each other.  I don't see how any modeling is done to reflect the clustering algorithms uncertainty of the underlying classes of each datapoint as alluded to in Definition 2.1.

---

> > > > > ### Author Response · Authors · 2023-11-23
> > > > >
> > > > > Thank you for your feedback. You mentioned a perceived gap between the definition of the problem and the introduction of probability in the reading, specifically in Definitions 2.1 and 2.2, where there is no mention of probability models. In response, we would like to clarify:
> > > > >
> > > > > - Definition 2.1 establishes the problem of interest, providing a general overview that is not inherently probabilistic but rather a general problem.
> > > > >
> > > > > - Definition 2.2 sets a goal, and immediately following Theorem 1 explains how to reduce current clustering uncertainty (i.e., increase NMI) by optimizing this goal. Specifically, the uncertainty reduces to 0 when the NMI reaches 1. The optimization involves selecting cluster pairs that are most likely to merge and lead to a great improvement in NMI. Previous work mostly considers the distance between the centroid samples of two clusters, which neglects many factors such as the overall sample distribution of two clusters and the purity of clusters. We attempt to solve these limitations by measuring the merge probability and the impact of merging clusters.
> > > > >
> > > > > - In Section 3.1, The paper introduces probability models to measure the specified metric, the expected improvement of Normalized Mutual Information (NMI), and the design of a probabilistic query strategy that selects the cluster pairs that can maximize the metric.
> > > > >
> > > > > We hope this clarifies the integration of probability models in the paper and its connection to the defined problems. In addition, we will further revise Section 2 to help readers understand and accept why we introduce probability and design the APC framework.

---

### Official Review · Reviewer_62ZR · 2023-10-31

**Soundness:** 3 good
**Presentation:** 1 poor
**Contribution:** 2 fair
**Rating:** 3
**Confidence:** 4

**Summary:**

The paper suggests a query-based clustering algorithm. The queries are: "are two points in the same target cluster." Human intervention is used to be able to answer such queries. The goal is to get close to a target clustering while making a small number of such queries. The paper suggests ideas to start with clustering and improve it by using queries to fuse and split clusters. Certain theoretical conditions are given for effective cluster fusion, and experimental results are given to show the utility of the suggested clustering ideas.

**Strengths:**

Query-based clustering is a relevant topic in the theory of clustering.

**Weaknesses:**

- The paper is not easy to read and understand. Even though the paper's main contribution is an algorithm, a clear description of the algorithm is missing in the main write-up. Multiple aspects of the algorithm have been deferred to the Appendix, and the writeup keeps pointing to the Appendix. For instance, consider the description of Algorithm 1 -- Algorithms 2 and 3 are deferred to the Appendix without giving the intuition regarding what they do. It is unclear what "Implement Human Test on w_1 and w_2" means.
- Theorem 2 gives some conditions under which cluster fusion gives an improvement. Are there reasons to believe such conditions could hold in natural clustering settings? Can these conditions be tested? Do these conditions continue to hold after a sequence of fusion operations?
- Does the initial clustering algorithm (FPC) use any queries? Is there some reason to believe that the initial clustering has some correlation with the target clustering? If so, what is the correlation, and how does this impact the number of queries? If not, what does FPC help? what if the target clustering is an arbitrary partition of the dataset and has nothing to do with geometric clustering ideas that place closer points in the same cluster? If the target clustering is an arbitrary partition of the dataset, what are the number of queries required to cluster?

With a lack of discussion on various issues and a lack of clarity on the suggested algorithm, it is difficult to form an informed opinion about the paper. The write-up should be improved to enable a fair review of the paper.

**Questions:**

Some of the questions are mentioned in the weakness section.

---

> ### Author Response · Authors · 2023-11-19
>
> Thank you for your and the other reviewers' valuable feedback. Taking into consideration the concerns raised regarding terminology, problem definitions, and algorithmic details, we have diligently revised the corresponding sections and added a flowchart illustrating how APC operates. We hope our response and the updated manuscript address your concerns and related issues about the paper. We answer your questions as follows:
>
> <Weakness 1> *Description of the main algorithm: Algorithms 2 and 3 are deferred to the Appendix without giving the intuition regarding what they do. What does "Implement Human Test on w_1 and w_2" mean?*
>
> Thanks for your advice. We placed Algorithms 2 and 3 in the main write-up instead of in the appendix in the revision. The intuition regarding them has been discussed in Section 3.3, and we rephrase them to help with your understanding in the revision.
>
> The "Human Test" consists of two parts: Purity Test and comparison of central samples. To avoid ambiguity in "Implement Human Test on w_1 and w_2", we have changed it to "Implement Purity Test on w_1 and w_2".
>
>
> <Weakness 2> *Can conditions in Theorem 2 hold in natural clustering settings? Do these conditions continue to hold after a sequence of fusion operations?*
>
> The conditions hold in some real scenarios such as face clustering and person re-identification, where FPC algorithm can provide a good initial clustering result and is widely applied in the real world [1]. We agree that it may not hold in some challenging tasks where the initial NMI is bad. However, it is challenging for a clustering algorithm to fit all datasets. Concerning the cases where the initial NMI is bad (e.g., 0.1), there are related works such as Active Deep clustering [2] aiming to solve it.
>
> Theorem 2 assures that the NMI value continues to increase after the lawful fusion operations (both conditions are satisfied), and the purity constraints are always tested before each fusion operation. Therefore, these conditions will continue to hold in APC.
>
> <Weakness 3>
> Q1: *Does FPC use any queries?*
>
> A1: FPC is an unsupervised clustering algorithm, and does not use human queries.
>
> Q2: *Is there some reason to believe that the initial clustering has some correlation with the target clustering? If so, what is the correlation, and how does this impact the number of queries?*
>
> A2: We contend that the clustering outcome of the common clustering method will exhibit a positive correlation with the target clustering, indicating accurate clustering for the majority of samples. For the specific scenarios in which the initial clustering does not correlate with the target clustering, we remark that the initial clustering is of low quality (low NMI and ARI value), and both FPC and APC are not proposed for such cases.
>
>
> The impact of the correlation is that the number of queries will decrease when the correlation is stronger. This is because a stronger correlation indicates a better initial clustering: clusters with high purity and clustering result with low category fission rate.
>
> [1] Liu etc. Mpc: Multi-view probabilistic clustering. CVPR (2022).
>
> [2] Bicheng Sun, Peng Zhou, etc. (2022). Active deep image clustering. Knowledge-Based Systems.

---

> > ### Comment · Reviewer_62ZR · 2023-11-22
> >
> > I want to thank the authors for their response. I have carefully considered the response and still think the paper will benefit from improved presentation. Given this, I would like to retain my initial score for the paper.

---

> > > ### Author Response · Authors · 2023-11-22
> > >
> > > Thank you for your response. We have outlined the revision in the general response. Any additional specific suggestions you may have regarding the writing and communication would be greatly appreciated.

---

### Official Review · Reviewer_2mue · 2023-10-31

**Soundness:** 2 fair
**Presentation:** 1 poor
**Contribution:** 2 fair
**Rating:** 3
**Confidence:** 4

**Summary:**

This paper considers the problem of active clustering. Pairs of points are given to a human to label whether or not the pair belongs to the same or different clusters. This paper uses an approximation to expected improvement of NMI to select pairs of clusters in the current clustering to use in the human query. Then representative samples from the pair of clusters is given to the human to label as same cluster or not. The human provided constraints are then incorporated by the clustering algorithm to improve the clustering. The empirical performance of the algorithm is shown on open source image datasets.

**Strengths:**

* The empirical results are very strong in comparison to the baselines
* The use of approximate expected NMI improvement to select queries for human labeling seems to be novel and intuitive.

**Weaknesses:**

* The writing needs some work in places. The authors use essential terminology and notation without definition: e.g. dominant class, purity, etc. There are a few minor typos as well throughout.
* There are some essential missing details of the method in Section 3. What exactly is the human answering in the Human Test? The lack of details in this section make it difficult to fully understand the proposed method
* It is unclear if the method is fair and could be applied in the real world. What information does the algorithm have access to? It seems like the method might have access to ground truth information and this is the reason it is performing so well.

**Questions:**

* Why is NMI of 0.95 enough? Is it possible in some applications that we would want say NMI of 1.0? Why not extend the results?
* What is a practical application in which we might want to utilize this method?
* Should the term *clustering* be used instead of *cluster* in Definition 2.1 and throughout the rest of the paper?
* What is the clustering algorithm used? The authors state the they use FPC, but how does this algorithm work?
* How are the constraints enforced by the clustering algorithm over time?

---

> ### Author Response · Authors · 2023-11-19
>
> Thank you for your and the other reviewers' valuable feedback. Taking into consideration the concerns raised regarding terminology, problem definitions, and algorithmic details, we have diligently revised the corresponding sections and added a flowchart illustrating how APC operates. We hope our response and the updated manuscript address your concerns and related issues about the paper. We answer your questions as follows:
>
> <Weakness 1> *The terminology like dominant class and purity is not clearly defined.*
>
> Assuming that the clustering of $N$ samples is $\Omega=\{w_1,\cdots,w_k\}$, and the ground truth clustering is $C=\{c_1,\cdots,c_K\}$. Then we define the dominant class of a cluster $w_i$ as $\arg \max_j |w_i\cap c_j|$, and the purity of $w_i$ as $\max_j \frac{|w_i\cap c_j|}{|w_i|}$. We have added the formal definition in Theorem 1 in the revision.
>
> <Weakness 2> *What exactly is the human answering in the Human Test?*
>
> In APC, the answer of the human query is either "must-link: the two samples belong to the same class" or "cannot-link: the two samples belong to different classes".
>
> <Weakness 3, Question 2> *What's the application of APC? What information does the algorithm have access to? Does it have access to the ground truth information which causes its good performance?*
>
> APC finds application in open-ended tasks like face clustering and person re-identification, where the number of classes is large and unknown. Evaluating the clustering quality and obtaining a reliable dataset in such contexts necessitates the input of human annotators. Their judgment becomes crucial in determining whether certain clusters genuinely represent the same person and warrant merging for an accurate and meaningful dataset.
>
> The APC algorithm has access to three pieces of information: 1) the clustering result of FPC; 2) the pairwise probability matrix $P_{N\times N}$; and 3) Human queries. We argue that APC does not require any ground truth information and its success is owing to two reasons: First, the FPC algorithm isolates noise samples into outlier clusters, leading to high purity in the remaining clusters. Second, APC can alleviate the category fission problem in FPC very fast.
>
> <Question 1> *Why is an NMI of 0.95 enough? Is it possible in some applications that we would want say NMI of 1.0? Why not extend the results?*
>
> Reaching an NMI of 1.0 with ACC methods proves challenging due to the difficulty in accurately classifying all noise samples. Our goal, however, is to significantly improve clustering quality with a practical and manageable number of human queries. Very few previous ACC methods extend it to 1.0, and it is beyond our problem setting.
>
> <Question 3> *Should the term clustering be used instead of cluster in Definition 2.1 and throughout the rest of the paper?*
>
> Thanks for pointing out this. We have revised it in the revision.
>
> <Question 4> *What is the clustering algorithm used? The authors state that they use FPC, but how does this algorithm work?*
>
> As outlined in Algorithm 1 (APC), FPC serves to produce the initial clustering result which is employed as APC's input. To better motivate the use of FPC, We added the description of FPC in the last paragraph of Section 3 in the revision.
>
> <Question 5> *How are the constraints enforced by APC?*
>
> We propose a relabeling strategy in Section 3.3, which directly adjusts the sample labels in the recalled cluster pair according to the constraints.
>
>
> [1] Mpc: Multi-view probabilistic clustering. CVPR 2022. Liu etc.

---

> > ### Comment · Reviewer_2mue · 2023-11-21
> >
> > Thank you for your response. Even after the edits to the paper, I think that the writing and communication needs to be improved quite a bit before publication.

---

> ### Author Response · Authors · 2023-11-22
>
> Thank you for your response. We have conducted a comprehensive revision, as outlined in our general response. Any additional specific suggestions you may have regarding the writing and communication would be greatly appreciated. We would be grateful if you could reconsider the assigned score. Your reconsideration would be highly valued.

---

### Author Response · Authors · 2023-11-22

Based on the valuable feedback from all reviewers, we have uploaded a revised version of our paper to address the concerns. The key modifications are summarized below:

- We have added a figure that shows the workflow of APC in Section 3 (i.e., Figure 1.).
- In Section 2, we have reorganized the comparison discussion between APC and other ACC methods regarding three important properties of ACC. On this basis, we clarify the motivation and objective of APC, which pursues a more effective and feasible cluster-based ACC framework.
- In Section 3.1, we have clarified the notation and meaning of the random variables, events, and conditional merge probability, and revised the relevant part in Appendix A.2.
- We have moved Algorithms 2 and 3 from the appendix to the main write-up in Section 3 and completed the missing details mentioned in the review.
- Various minor modifications have been implemented based on the suggestions from the reviewers.

All changes in this revision are highlighted in blue. We hope that these revisions effectively address the concerns raised by the reviewers. Should there be any additional questions from the reviewers, please do not hesitate to inform us, and we will be delighted to address them.